# Vancomycin-Loaded 3D-Printed Polylactic Acid–Hydroxyapatite Scaffolds for Bone Tissue Engineering

**DOI:** 10.3390/polym15214250

**Published:** 2023-10-28

**Authors:** Sara Pérez-Davila, Carmen Potel-Alvarellos, Raquel Carballo, Laura González-Rodríguez, Miriam López-Álvarez, Julia Serra, Patricia Díaz-Rodríguez, Mariana Landín, Pío González

**Affiliations:** 1CINTECX, Universidade de Vigo, Grupo Novos Materiais, 36310 Vigo, Spainmiriammsd@uvigo.gal (M.L.-Á.);; 2Galicia Sur Health Research Institute (IIS Galicia Sur), SERGAS-UVIGO, 36213 Vigo, Spain; carmen.potel.alvarellos@sergas.es (C.P.-A.);; 3Laboratorio de Microbiología, Complejo Hospitalario Universitario de Vigo, 36312 Vigo, Spain; 4Pharmacology, Pharmacy, and Pharmaceutical Technology Department, I+D Farma (GI-1645), Faculty of Pharmacy, Institute of Materials, iMATUS and Health Research Institute of Santiago de Compositela (IDIS), University of Santiago de Compostela, 15705 Santiago de Compostela, Spain; patricia.diaz.rodriguez@usc.es (P.D.-R.); m.landin@usc.es (M.L.)

**Keywords:** polylactic acid, hydroxyapatite, vancomycin, 3D printing, antibacterial, biocompatibility

## Abstract

The regeneration of bone remains one of the main challenges in the biomedical field, with the need to provide more personalized and multifunctional solutions. The other persistent challenge is related to the local prevention of infections after implantation surgery. To fulfill the first one and provide customized scaffolds with complex geometries, 3D printing is being investigated, with polylactic acid (PLA) as the biomaterial mostly used, given its thermoplastic properties. The 3D printing of PLA in combination with hydroxyapatite (HA) is also under research, to mimic the native mechanical and biological properties, providing more functional scaffolds. Finally, to fulfill the second one, antibacterial drugs locally incorporated into biodegradable scaffolds are also under investigation. This work aims to develop vancomycin-loaded 3D-printed PLA–HA scaffolds offering a dual functionality: local prevention of infections and personalized biodegradable scaffolds with osseointegrative properties. For this, the antibacterial drug vancomycin was incorporated into 3D-printed PLA–HA scaffolds using three loading methodologies: (1) dip coating, (2) drop coating, and (3) direct incorporation in the 3D printing with PLA and HA. A systematic characterization was performed, including release kinetics, *Staphylococcus aureus* antibacterial/antibiofilm activities and cytocompatibility. The results demonstrated the feasibility of the vancomycin-loaded 3D-printed PLA–HA scaffolds as drug-releasing vehicles with significant antibacterial effects for the three methodologies. In relation to the drug release kinetics, the (1) dip- and (2) drop-coating methodologies achieved burst release (first 60 min) of around 80–90% of the loaded vancomycin, followed by a slower release of the remaining drug for up to 48 h, while the (3) 3D printing presented an extended release beyond 7 days as the polymer degraded. The cytocompatibility of the vancomycin-loaded scaffolds was also confirmed.

## 1. Introduction

Implantable bone-related biomaterials are essential devices that have revolutionized the quality of life for millions of people worldwide. The continuous advances in materials science in manufacturing techniques and in the understanding of the biological response to implantation have enhanced their safety and functionality [1]. However, the regeneration of damaged bone tissue remains one of the main challenges in the biomedical field, especially concerning the provision of biodegradable, personalized and multifunctional solutions. These new general perspectives have led polymers becoming the biomaterials that dominate the engineering of soft tissues and the drug delivery industry, while in hard tissue engineering, that related to bone tissue, polymers are also gradually replacing metals by their combination with bioceramics to mimic some of the native bone mechanical and biological properties [2].

In relation to these bone-related scaffolds, new requirements based on the ability to manufacture scaffolds in irregular shapes, as well as their multifunctional role, are then added to the already well-known critical parameters such as porosity (size, volume, interconnectivity) and mechanical strength [3,4]. In addition, the use of biodegradable polymer-based scaffolds offers the advantage of gradual replacement by new tissue during the degradation process, eliminating the need for additional interventions to remove the biomaterial from the body [5]. Among all the biocompatible and biodegradable polymers, polylactic acid (PLA) stands out, which is also biodegradable and already widely used in clinics and accepted by the US Food and Drug Administration (FDA) in almost all medical specialties, such as medical implants, porous scaffolds and drug delivery systems [6,7,8].

Recently, the interest in PLA has dramatically increased due to its thermoplastic properties, making it one of the most extensively utilized filaments in emerging 3D-printing technologies like fused deposition modeling (FDM). The combination of this fabrication technology with PLA properties will allow the rapid fabrication of customized biodegradable scaffolds based on clinical images of the patient and with complex geometries and controlled architectures [9,10,11]. Full control of pore size and interconnectivity is guaranteed in comparison to certain limitations offered by fabrication methods, such as chemical/gas foaming, solvent casting, particle/salt leaching, freeze-drying, thermally induced phase separation, foam gel, electrospinning and stereolithography [12].

Finally, in order to mimic composition, and thereby get closer to the native mechanical/biological properties of the bone tissue, the 3D printing of PLA in combination with bioceramics such as hydroxyapatite (HA) is also under research to provide the scaffolds with osteoconductive, osteoinductive [13,14,15] and osteogenic properties [16] as well as improved stiffness and the partial counteract of the pH drop [17]. In this aspect, the obtaining of 3D-printed scaffolds from direct mixtures of PLA and HA particles, avoiding the need for a prefabricated filament, has been recently demonstrated [18]. In this previous work, scaffolds with HA contribution up to 13 wt.% were validated, with modulated ranges in pore size from 250 to 850 µm and porosity from 24% to 76% in total volume. This versatile printing methodology opens the possibility to directly obtain, even in the operating room itself, personalized modulated compositions with porous structures within the ranges reported in the literature to promote bone formation and vascularization.

The other pending challenge in the biomedical field is related to the local prevention of infections after surgery performed for the biomaterial implantation, as they can cause morbidity, even mortality, and present high economic costs [19]. Minimizing the impact of these infections is crucial, as the implantation of biomaterials is expected to increase in the coming years. Concerning bone-related infections, several strains of Gram-positive *Staphylococcus aureus* and associated biofilms are responsible for a significant proportion, originating mainly from the patient’s own skin, the health-care personnel, or the environment [20]. These infections can induce severe bone loss and destruction resulting from an imbalance between osteoblasts and osteoclasts, and are characterized by recurrence and even resistance to antibiotics and antimicrobial agents in several strains [21,22,23,24]. To address this, the local loading of antibiotics into porous biodegradable bone-related scaffolds to treat/prevent these infections nowadays is a common research strategy, and vancomycin is a promising agent that exhibits minimal adverse effects on osteoblast and bone regeneration. These drug delivery scaffolds will be able to release an adequate local concentration of the antibacterial agent for therapeutic or preventive purposes, and at the same time reduce the systemic toxicity and side effects of parenteral antibiotics [24,25,26]. Among all the methodologies investigated for the incorporation of antibacterial agents in the 3D constructs, 3D printing is being seriously considered [27].

Thus, the present study aimed to develop vancomycin-loaded 3D-printed PLA–HA scaffolds offering a dual functionality: local prevention of infections and personalized 3D constructs for bone regeneration. For this, the antibacterial drug vancomycin was incorporated into the scaffolds using three loading methodologies: (1) dip coating of the 3D-printed PLA–HA scaffold, (2) drop coating, and (3) direct incorporation, along with the PLA and HA particles, during the 3D printing. A diagram with the proposed strategy based on the direct obtaining of the personalized loaded scaffolds in the operating room for clinical application is presented in Figure 1. A systematic characterization is then provided, beginning with the determination of the vancomycin release kinetics for each loading methodology, together with microbiological assays to evaluate antibacterial efficacy against *S. aureus* biofilm. Agar diffusion and direct contact tests on the scaffold surface were carried out. Antibiofilm activity including quantification of live/dead bacteria (ddPCR) was also determined. The potential cytotoxicity of the loaded scaffolds was analyzed in vitro with both NCTC clone 929 fibroblasts and MG63 osteoblast-like cell lines, including solvent extraction tests, cell morphology (SEM) and direct cell proliferation (MTS assay) up to 6 days.

## 2. Materials and Methods

### 2.1. 3D Printing and Vancomycin Loading

Natural polylactic acid (PLA) SMARTFIL^®^ in oval-shaped pellets, with dimensions of 5 × 3.5 mm^3^, were purchased from Smart Materials, Jaén, Spain (technical data of PLA SMARTFIL^®^ provided in Appendix A). For hydroxyapatite (HA), we used hydroxylapatite Captal^®^ “R” (batch P120R) powder in spherical morphology (average particle size 3.29 µm), acquired from Plasma-Biotal Limited (Derbyshire, UK). According to the manufacturers, it presents a Ca:P ratio in the range 1.66–1.72, crystallinity around 85–95% and a high surface area of typically 6–20 m^2^/g (http://www.plasma-biotal.com/captal-r-hydroxylapatite/, 28 September 2023).

Homogeneous mixtures (10 g) composed of PLA (97 wt.%) and HA (3 wt.%) were prepared in a petri dish and directly introduced in a 3D FDM printer hopper (TUMAKER Voladora NX Pellet, Valencia, Spain) to obtain 3D-printed PLA–HA scaffolds shaped as disks, with dimensions of 12 mm diameter and 3 mm height and an infill density of 90%. These methodology has been extensively described and optimized in a previous article [14].

Once fabricated, the scaffolds described above were loaded with the antibiotic vancomycin (vancomycin hydrochloride; lyophilized powder, Pfizer, New York, NY, USA, 3157156), following two methodologies: (1) dip coating and (2) drop coating. First, in the case of (1) vancomycin dip coating, the 3D-printed PLA–HA scaffolds were immersed in 1 mL of 42.5 mg/mL vancomycin solution in ultrapure water for 1 h in a vacuum chamber to allow the solution to penetrate the scaffold pores. Then, the dip-coated scaffolds were dried at 37 °C for 24 h. To obtain the (2) vancomycin drop-coated PLA–HA scaffolds, a drop of 200 µL of vancomycin solution of 6.25 mg/mL in ultrapure water (1.25 mg of vancomycin) was added to the surface of each 3D-printed PLA–HA scaffold. These drop-coated scaffolds were also dried at 37 °C for 24 h.

Finally, the direct incorporation of vancomycin along with PLA and HA during the (3) 3D printing constitutes the third methodology evaluated. Those 3D-printed scaffolds were prepared by homogeneously mixing vancomycin (2 wt.%) in solid form (lyophilized powder) with HA (3 wt.%) and PLA (95 wt.%) in a petri dish and directly introducing the mixture in the 3D FDM printer.

### 2.2. Vancomycin Release Studies

The vancomycin-loaded 3D-printed PLA–HA scaffolds obtained by the three methodologies were transferred to vials containing 5 mL of phosphate-buffered saline (PBS) at pH 7.4 and 37 °C and maintained under mechanical stirring. At different time points, samples of the release medium (2 mL) were withdrawn and replaced with fresh PBS. The absorbance of each aliquot was evaluated in a UV spectrophotometer at 280 nm (Agilent 8453, Santa Clara, CA, USA) and the vancomycin concentrations calculated using a vancomycin calibration curve. Unloaded 3D-printed PLA–HA scaffolds were used as controls to remove the effect of degradation products on the absorbance signal. Data are expressed as percentages of cumulative drug released.

### 2.3. Solvent Extraction Cytotoxicity, Cell Morphology and Proliferation

Biological assays were then performed to evaluate the mammal cell lines’ response to the vancomycin-loaded scaffolds. To obtain the scaffolds to be used in these assays, the previously described 3D-printing process was performed in aseptic conditions and under a UV lamp situated over the printer. The printing process, carried out over −200 °C, can be considered a sterilization technique [9]. The scaffolds were exposed (both sides) to UV radiation in a laminar flow cabinet for 30 min per side to avoid potential contamination.

First, the potential cytotoxicity caused by the released particles/chemicals/reagents from the different vancomycin-loaded scaffolds was evaluated in a solvent extraction test, carried out following the indications of UNE-EN-ISO 10993-5:2009 [33] with the cell line NCTC clone 929 (ECACC 88102702) using mouse fibroblasts. To prepare the extracts, the scaffolds were first placed in individual Falcon tubes with the culture medium DMEM (Lonza, Basilea, Switzerland) supplemented with 10% of fetal bovine serum (HyClone Laboratories LLC, Logan, UT, USA) and 1% of a combination of penicillin, streptomycin. and amphotericin B (Lonza, Basilea, Switzerland) in a ratio of 3 cm^2^ of material per mL of medium (UNE-EN-ISO 10993-12:2021) [34] and kept at 37 °C for 24 h with 60 rpm shaking. After that, different concentrations (100%, 50%, 30%, 10% and 0%) of each extract were prepared by diluting the initial ones with fresh culture medium. A 6.4 g/L phenol solution was used as a positive control, and the supplemented culture medium itself was also subjected to the same extraction process. Simultaneously, a suspension of 1 × 10^5^ cells/mL was seeded in a 96-well microplate for 72 h of incubation. At that time, the cell medium was replaced by the different concentrations of the extracts and cells were incubated for 24 h (four replicates per concentration). Afterwards, the absorbance was quantified using an MTS Cell Proliferation Assay Kit (Abcam, Cambridge, UK). The MTS reagent was added to each well and incubated for 45 min. The absorbance was measured at a wavelength of 490 nm in a microplate spectrophotometer (Bio-Rad, Hercules, CA, USA). All experiments included a negative control of cytotoxicity, the cell culture medium, and a positive control, a concentrated phenol solution. Results are expressed as the percentage of viability compared to the negative control of cytotoxicity ± standard error, and the dotted line indicates the limit (70%) of cytotoxicity according to [33].

Direct seeding of the cells onto the loaded scaffolds was also performed to evaluate cell morphology and proliferation. Vancomycin-loaded 3D-printed PLA–HA scaffolds obtained by the three loading methodologies were placed in 48-well microplates and seeded with MG63 human osteosarcoma cells (ECACC, Salisbury, UK) of 7 × 10^4^ cells/mL in 300 µL of EMEM (Lonza, Basilea, Switzerland), supplemented with 10% fetal bovine serum (HyClone Laboratories LLC, Logan, UT, USA) and 1% of a combination of penicillin, streptomycin and amphotericin B (Lonza, Basilea, Switzerland). Unloaded scaffolds were used as negative control. Empty tissue culture polystyrene (TCP) microplate wells were also seeded with the same cell suspension to be used as the gold standard to confirm the health of cells. After 6 days of incubation, the cells on the scaffolds were washed with PBS (phosphate-buffered saline, Lonza, Basilea, Switzerland) and fixed with 2.5% glutaraldehyde (Sigma Aldrich, St. Louis, MI, USA) in PBS for 2 h at 4 °C. The cells seeded on the scaffolds were then washed three times for 30 min each with PBS and dehydrated in graded acetone solutions for 30 min in each solution and in absolute acetone for 1 h. After dehydration, the samples were submitted to a critical point in CO_2_, at 75 atm and 31.3 °C as a final step, mounted on metal stubs and sputter-coated with gold prior to their analysis using a Quanta200 scanning electron microscope (CACTI, University of Vigo, Vigo, Spain). Finally, cell proliferation was quantified after the same period of incubation (6 days), again using the MTS Cell Proliferation Assay Kit (Abcam, Cambridge, UK). For this, the MTS reagent was added to the corresponding wells and the absorbance values were determined at 490 nm in a microplate spectrophotometer (Bio-Rad, Hercules, CA, USA). Results are expressed as the percentage of viability compared to the negative control of cytotoxicity ± standard error.

### 2.4. Antibacterial (Diffusion, Direct Contact) and Antibiofilm Activity

Antimicrobial activity analysis was carried out using the *Staphylococcus aureus* strain of reference—ATCC 25923. Bacteria were cultured on blood agar plates (TSS, bioMérieux España SA, Madrid, Spain) and at the same time two bacterial colonies were transferred to 5 mL of brain heart infusion broth (BHI, bioMérieux España SA, Madrid, Spain) and incubated for 18 h at 37 °C. For all the methods studied, three replicates of the loaded scaffolds per each methodology were used, including the controls: one negative with the corresponding loaded scaffolds and growth medium without *S. aureus* and one positive with the unloaded scaffolds.

Initially, the antibacterial activity was evaluated using the Agar disk diffusion test (diffusion method). The bacterial suspension was adjusted to 0.5 McFarland units, which is equivalent to an optical density comparable to the density of a bacterial suspension with 1.5 × 10^8^ colony-forming units per mL (CFU/mL), and used to inoculate Mueller–Hinton agar (MHA, bioMérieux España SA, Madrid, Spain) plates. After 5 h of drying, the vancomycin-loaded disks were placed in the center of the inoculated plates, incubated for 24 h at 37 °C, and the inhibition halos measured. Secondly, the antibacterial activity was analyzed on the material’s surface (contact method). This method corresponds to that followed by Martí et al. for the measurement of antimicrobial activity on plastic surfaces [35]. The bacterial suspension was adjusted to 0.5 McFarland and diluted to 10^6^ CFU/mL. The disks were placed in 12-well plates, and 150 µL of the bacterial suspension was placed on the surface of each disk and incubated for 24 h at 37 °C. Then, 3 mL of BHI was added, mixed well, and ultrasound applied at 50 Hz for 5 min. Finally, the standard plate-counting method was carried out, which consists of diluting the sample with sterile saline solution until the bacteria are sufficiently diluted to count the CFU/mL accurately. The percentage of viability loss (LV) was calculated from the colony-counting results as the number of CFU/mL·cm^2^ from unloaded samples (C) and loaded samples after 24 h of culture (S), in accordance with Equation (1):(1)$$LV\left(\%\right)=\frac{C-S}{C}\times 100$$

Finally, the antibiofilm activity was also evaluated by the incubation of the loaded scaffolds with 2 mL of a bacterial suspension at 1.5 × 10^7^ CFU/mL in 12-well plates for 48 h, with a renewal of the culture medium after 24 h. Afterwards, the BHI was removed and the disks were transferred to clean wells, washed 3 times with PBS and sonicated with 2 mL of PBS for 5 min at 50 Hz. Standard plate-counting data are represented as log (CFU/cm^2^). Moreover, 100 µL of the above biofilm suspensions was also collected for quantification of live and dead bacteria by droplet digital quantitative PCR (ddPCR) together with the propidium monoazide dye (PMA, Biotium, Fremont, CA, USA), which uses membrane integrity as a viability criterion [36]. After photolysis of this PMA, it is intercalated only into the DNA of dead cells, thus avoiding the major drawback of false positives in DNA replication of dead bacteria by conventional PCR. For this, the sample was divided in two (50 µL), leaving one part in a cryovial (untreated sample) and the other part to be treated with 2 µL of PMA at 2.5 µM to perform the photolysis (GloPlate TM Blue LED Illuminator, Biotium, Fremont, CA, USA) for 10 min at 4 °C in the dark. Then, 150 µL of sterile PBS was added, centrifuged, and the pellet resuspended in 200 µL of PBS. Both treated and untreated samples were frozen at −20 °C until the next step. Then, the samples previously treated with lysozyme (20 mg/mL in Tris-EDTA 1x buffer, PanReac Applichem, Barcelona, Spain) were extracted and purified using the QIAamp^®^ DNA mini kit (Qiagen, Hilden, Germany) according to the manufacturer’s instructions to finally perform the ddPCR. In this step, the samples were previously treated with a mix that included the FAM-labeled Taqman hydrolysis probe in order to proceed to the generation of droplets with the equipment (Automated Droplet Generation, Bio-Rad, Hercules, CA, USA). The amplification of each of the droplets was performed through a thermal cycler (Thermal Cycler C1000Tcuch™, Bio-Rad, Hercules, CA, USA) and a droplet reader (Droplet Reader QX200™, Bio-Rad, Hercules, CA, USA) and assessing the positive or negative fluorescence to calculate the concentration of target DNA in order to analyze it through the QuantaSoft™ Software, version 1.7 (Bio-Rad, Hercules, CA, USA).

### 2.5. Statistical Analysis

All biological data were analyzed using GraphPad Prism 8 (GraphPad Software Inc., San Diego, CA, USA). The nonparametric Mann–Whitney U test was used to determine the statistical differences between the control (unloaded) and the different loading methodologies. Statistical significance was determined to be * (*p* ≤ 0.05) at the 95% confidence level.

## 3. Results and Discussion

### 3.1. Vancomycin Release Kinetics

Figure 2A shows the vancomycin release profiles from the 3D-printed PLA–HA scaffolds loaded by the three different methodologies: (1) dip coating, (2) drop coating and (3) 3D printing. The drug release profiles were obtained in phosphate buffer at pH 7.4 to simulate the physiological release process. Differences were found depending on the loading methodology used. Thus, when the scaffolds were loaded by dip and drop coating, two-step release profiles were obtained. A burst release of 80–90% of the total amount of vancomycin was released during the first 60 min, followed by a slower drug release up to 100% in the next 6–7 h (critical period after a surgery). The high water solubility of vancomycin (>100 mg/mL) justifies the rapid initial release in the first hours detected for these loading methodologies [37], as the antibiotic is located on the external surface of both types of loaded scaffolds. On the contrary, scaffolds loaded using the 3D-printing methodology showed a burst release for the first hour of around 20% of vancomycin, followed by a controlled release of vancomycin, which can be related to the formation of a PLA–HA matrix system where the drug is homogeneously distributed. This matrix was able to control the release of the hydrophilic compound following a Higuchi kinetic profile (r = 0.9714) indicative of a diffusion controlled release mechanism.

The two different antibiotic release profiles, (i) fast release for dip and drop coating, and (ii) slow release for 3D printing, are of enormous interest to avoid bacterial colonization during the implantation or fixation of biomedical devices, such as arthroplasties, dental implant surgeries, etc. The first hours after surgery are crucial to avoid infectious agent colonization, preventing bacterial adhesion and potential biofilm formation. An initial fast release should help in reducing the development of drug resistance and fixation failure [37,38]. After this initial step, a sustained drug release is desirable to ensure a sufficient amount of antibiotics in the injury microenvironment to inhibit or eradicate potential bacterial remnants at the fixation site [39]. In this sense a combination of the different loading methodologies will also be applied in cases of interest. Moreover, based on the previously described data, the expected antibiotic concentration at the implantation site should guarantee a bactericidal effect at all time points, being higher than the minimum inhibitory concentration (MIC) for *S. aureus* [39].

### 3.2. Solvent Extraction Cytotoxicity, Cell Morphology and Viability

Figure 2B shows the results of the cytotoxicity assay carried out with the extracts of the different loaded scaffolds incubated with the mouse fibroblast cells (NCTC clone 929) in different concentrations from 0% to 100% (pure extract). As can be seen, cell viability values for all extracts were higher than 90%, above the cytotoxicity limit of 70% established by the standard [33]. Moreover, no statistically significant differences were found for cell viability values between all the extracts for the three methodologies compared to the negative control, neither at 100% nor those diluted at 50%, 30%, 10% and 0%. The statistically significant differences in cytotoxicity found between the positive control and the 100%, 50%, 30% and 10% extracts validated the experiment.

The results are in agreement with other authors [37] who recently used the same solvent extract test to evaluate the potential cytotoxicity of Schanz nail extracts coated with PLA and vancomycin. Moreover, a solution of vancomycin (500 μg/mL) with povidone iodine, designed to prevent biofilm formation of resistant *S. aureus* strains on biomaterials, was also recently proved as nontoxic for the skeletal muscle tissue [40].

Figure 2C shows the viability of MG63 cells (osteoblast-like cells) after being directly incubated for 6 days on the three types of vancomycin-loaded scaffolds, normalized with the values obtained for the unloaded 3D-printed PLA–HA scaffolds. As can be observed, cell viability percentages for loaded scaffolds were close to or even higher than the values obtained for the unloaded scaffolds (100%), indicating that the drug loading does not affect the viability.

Morphological characterization of the adhered cells by SEM allows us to evaluate the cell–material interaction (Figure 2D). At 200× magnification, images for the three types of loaded scaffold (a, c and e) show areas completely covered with a cell monolayer, indicating the proper adhesion and spreading of the cells. Dip coated scaffolds (a) additionally show the ingrowth of the cells into the pores. At 1600× magnification, a high density of healthy appearance cells with abundant intercellular contacts can be observed (Figure 2D(b,d,f)). Furthermore, in Figure 2D(d,f) (drop and 3D printing methodology), extended filopodia establishing direct contact with neighboring cells and with the scaffold surface are clearly presented, indicating good adhesion. These results of cell viability and proliferation agree with previous literature, such as the work published by Lian et al. [41]. These authors have demonstrated the in vitro biocompatibility of vancomycin-loaded nanohydroxyapatite–collagen–PLA composites with rabbit bone marrow stromal cells after a release rate of vancomycin of 0.9 mg/mL during 7 days of incubation.

### 3.3. Antibacterial and Antibiofilm Activity

The antibacterial activity was first evaluated with the agar diffusion method, incubating the vancomycin-loaded scaffolds for 24 h in the Mueller–Hinton agar plates previously inoculated with *Staphylococcus aureus*. Unloaded scaffolds were used as control. Figure 3A shows the inhibition halos for each scaffold type. As expected, no evidence of inhibition halo was observed in the control scaffold. Scaffolds loaded by either drop- or dip-coating methodologies showed similar inhibition halos of 28.7 ± 0.3 and 27.0 ± 0.6 mm in diameter, indicating similar antibacterial effect, which is in agreement with their similar and fast drug release profile. After 24 h, the inhibition halo observed for scaffolds loaded by the 3D-printing methodology was just 16.2 ± 1.3 mm in diameter, which corresponds with a much slower release rate (Figure 1). Despite the smaller size of the inhibition zone, the drug load, and the release rate, vancomycin-loaded 3D-printed scaffolds are sufficient to guarantee antibacterial effect. The particular dose of vancomycin printed in the present work causes the antibacterial effect at short times to be lower than with the other two loading methodologies. In this regard, it is important to take into account that different doses of vancomycin have been incorporated with the three methodologies, and therefore a quantitative comparison in terms of antibacterial activity between the methodologies cannot be given. The results indicate that the vancomycin-loaded scaffold using 3D printing maintained its properties as an antibacterial agent. Vancomycin molecules remained stable, despite the 3D-printing processing temperatures (up to 240 °C), including the ultraviolet radiation used for sterilization (15 min during processing plus 30 min in laminar flow chamber, 45 min in total).

These data agree with the study by Ranganathan et al., where the effect of temperature and exposure to ultraviolet light on the effectiveness of different antibiotics, including vancomycin, was evaluated. Their results indicated that even after exposure to different temperatures between 20 °C and 230 °C or to ultraviolet light at 400 nm for short periods (10 min), vancomycin does not lose its effectiveness (inhibition ratios present in *E. coli* cultures) [42]. Finally, the dip-coating results also agree with a previous work [43]. However, the inhibition halos obtained by these authors were smaller, undoubtedly due to the lower concentration of vancomycin used (5 mg/mL) compared to the present work (42.5 mg/mL).

The antibacterial activity was also evaluated by the contact method with the *S. aureus* inoculum after 24 h of incubation. Figure 3B presents the colony counts, directly and after serial dilutions, corresponding to the assay with unloaded and vancomycin-loaded scaffolds by the 3D-printing methodology. As can be seen, there is a substantial disparity in bacterial growth for unloaded and loaded scaffolds. Unloaded scaffolds exhibited a complete coverage with colonies up to 10^−3^ dilution, while colonies are hardly detected, even at the initial (direct) dilution, in the loaded ones. This result clearly indicated a strong antibacterial activity due to the almost total inhibition on the growth of *S. aureus* by direct contact with the antibiotic vancomycin.

Quantitative antibacterial activity was assessed using the parameter loss of viability (LV%) (Equation (1)) for the different scaffolds studied. The results (Figure 3C) confirmed that the three types of vancomycin-loaded scaffolds were able to inhibit the growth of *S. aureus* by the direct contact method, showing positive antibacterial results. The loss of viability was 99.99% for scaffolds loaded by 3D printing (images presented in Figure 3) and 100% for the other two methodologies (dip and drop coating), with no significant differences between them.

Finally, the antibiofilm properties of the vancomycin-loaded scaffolds were also evaluated. Figure 4A shows the images of the biofilm formation experiment that was carried out with the three types of scaffolds, incubating them for 48 h into 2 mL of *S. aureus* suspension. Unloaded scaffolds were included as controls. After this time, unloaded scaffolds showed a visible gelatinous layer clouding the liquid. Vancomycin-loaded scaffolds showed a complete absence of turbidity, indicating none or a reduced number of microorganisms; therefore, all of them present a certain degree of antibiofilm effect independently of the method used for their loading. The quantification of colony-forming units was then carried out, and the data presented in log reduction of CFU/cm^2^ are shown in Figure 4B. These data indicated a significant reduction (*p* < 0.05) for the vancomycin-loaded scaffolds of the three methodologies with respect to the control. Scaffolds processed by 3D printing, which incorporated 2 wt.% vancomycin, achieved a logarithmic reduction of 1.63, lower than those loaded by dip coating (3.21) or drop coating (3.10), which can be related to the amount of vancomycin loaded, together with the drug release properties. Considering the vancomycin release kinetics for each methodology and the antibiotic amounts loaded, it is reasonable that scaffolds processed by dip and drop coating promoted in the short term a reduction of >99.9% of the viable number of CFUs after 48 h of incubation time. The scaffolds with vancomycin loaded by 3D printing promoted a reduction of 90% to 99% of the viable number of CFUs.

Figure 4C shows dead and live bacterial cells expressed as log reductions of copies per mL (cp/mL). As can be seen, the highest number of total bacteria (dead + live) was found in the unloaded scaffolds. The scaffolds loaded by dip- and drop-coating methodologies showed the lowest number of total bacteria and of live bacteria (almost cero). Scaffolds loaded by 3D-printing methodology showed intermediate results. They presented a lower number of live bacteria than the control, preventing their exponential growth. The number of dead bacteria for the unloaded scaffolds was unexpectedly high. This fact can be justified by the hydrophilic nature of these scaffolds (contact angle measurements provided in Appendix A), which can influence the bacterial adherence to the surface of the biomaterial [19]. Maikranz et al. showed that *S. aureus* adheres more strongly to hydrophobic than hydrophilic surfaces [44]. It may then be relevant to consider the hydrophilicity of the biomaterials used to manufacture the scaffolds, as they may play a key role in the initial adhesion of bacteria that help to prevent infections related to their implantation.

The three methodologies for vancomycin loading in 3D-printed PLA–HA scaffolds tested in the present work could be applied separately or in combination, depending on the requirements of the personalized scaffold applications for both the amount of drug and the release kinetics. The loading of vancomycin in titanium implant surfaces was recently tested by Taha et al. [40] with a dose of 500 µg/mL of vancomycin and tested for biofilm formation. Results showed antibiofilm properties of resistant *S. aureus* strains and the absence of toxicity to the skeletal muscle tissue. Li et al. [45] recently proposed the incorporation of vancomycin in poly(lactic-co-glycolic acid) microspheres in concentrations up to 200 µg/mL to then be loaded into the scaffolds. The present work proved the loading capacity of higher doses of vancomycin (as with the 6.25 mg/mL in drop coating) together with the absence of cytotoxicity along with antibacterial efficacy. In relation to the 3D-printing methodology, a previous work [38] demonstrated that vancomycin remains stable and effective after being subjected to temperatures between 20 to 230 °C and ultraviolet light at 400 nm for short times (10 min). The present work demonstrates that the stability and effectivity is also guaranteed up to 240 °C and up to 45 min of ultraviolet radiation exposure (as in the case of 3D printing) in total for sterilization. Moreover, despite the fact that the absence of cytotoxicity was confirmed, together with the antibacterial activity of the scaffolds loaded by the three methodologies, the physicochemical properties of the unloaded 3D-printed PLA–HA scaffolds were proven to present a certain degree of antibacterial activity (with an unexpectedly high number of dead bacteria) against *S. aureus*. In relation to this, several authors have investigated the influence of the surface of a biomaterial on the bacterial adherence [19], which in the case of *S. aureus* seems to be significantly higher on hydrophobic surfaces [44]. The hydrophilicity of the 3D-printed PLA–HA scaffolds proposed may play a key role in the initial adhesion of bacteria and should be considered as an added advantage for their local prevention of infections.

## 4. Conclusions

The work demonstrates the feasibility of developing vancomycin-loaded 3D-printed PLA–HA scaffolds offering a dual functionality: local prevention of infections against *S. aureus* and personalized biodegradable scaffolds with osseointegrative properties. The vancomycin release kinetics obtained for the three loading methodologies tested—(1) dip coating, (2) drop coating, and (3) direct incorporation of vancomycin during the 3D-printing process—proved the viability of the 3D-printed PLA–HA scaffolds as drug delivery vehicles for local controlled release. Moreover, the tested methodologies were proven to present different vancomycin release profiles, with an initial rapid release (60 min) of around the 89–90% of loaded vancomycin, followed by a slower liberation of that remaining during the next 48 h for dip and drop coating, and a much slower profile for 3D printing, extending the release period beyond 7 days as the polymer degrades.

Versatility is thus provided to the clinician, being able to modulate the vancomycin release from the scaffolds with a combination of the different methodologies. The loaded 3D-printed PLA–HA scaffolds for the three methodologies were proven to be effective for antibacterial and antibiofilm activity against *S. aureus*. The 240 °C achieved during the 3D-printing process and the 45 min of maximum ultraviolet radiation exposure of the corresponding loaded scaffolds for sterilization did not affect the stability or effectivity of the vancomycin. Finally, the absence of cytotoxicity was also confirmed with fibroblast- and osteoblast-like cells.

## Figures and Tables

**Figure 1 polymers-15-04250-f001:**
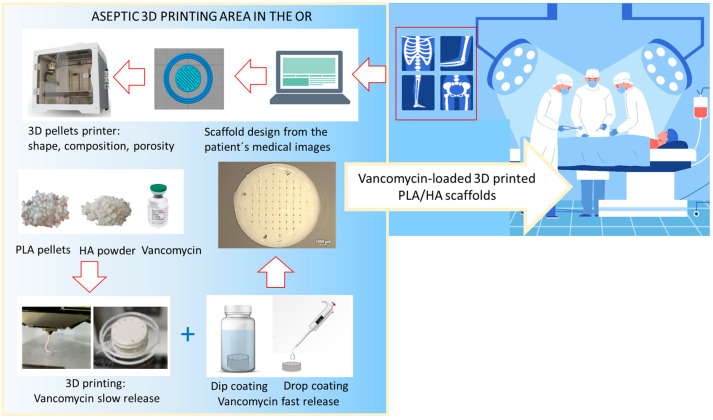
Schematic figure with the project’s idea and application. This figure has been created using images designed by Freepik [28,29,30,31,32].

**Figure 2 polymers-15-04250-f002:**
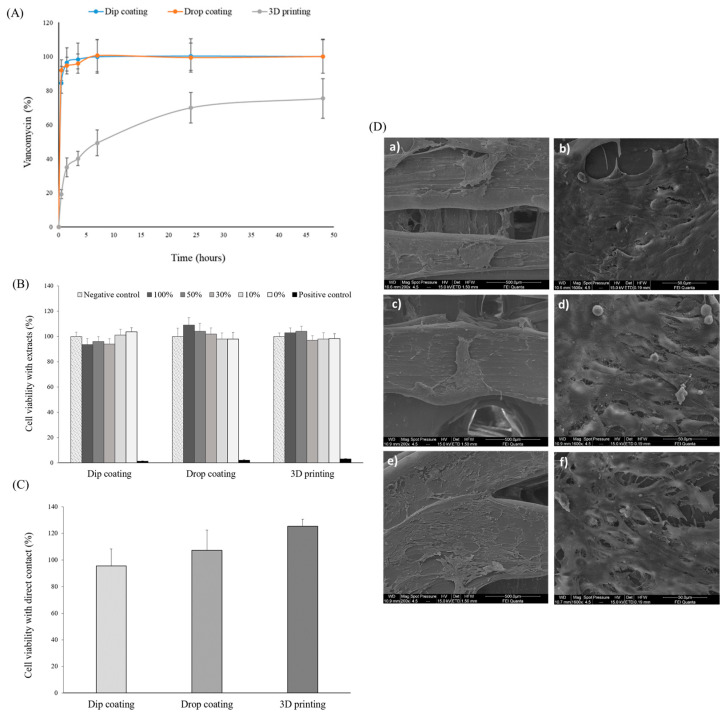
(**A**) Vancomycin release profiles, in phosphate buffer up to 48 h, from the 3D-printed PLA–HA scaffolds loaded by the three different methodologies: dip coating, drop coating and 3D printing. (**B**) Percentage of viability of the mouse fibroblast cells (NCTC clone 929) with the extracts of vancomycin-loaded 3D-printed PLA–HA scaffolds diluted at 100%, 50%, 30% and 10%. Results are expressed as percentages compared to the negative control ± standard error. The dotted line indicates the limit (70%) of cytotoxicity according to [33] and the statistical significance represented as * for *p* ≤ 0.05, (**C**) Cell viability detected after 6 days of incubation in MG63 human osteosarcoma cells in direct contact with the three vancomycin-loaded PLA–HA scaffolds. Results are expressed as percentage of cell viability compared to the metabolic activity quantified on the unloaded scaffolds, considered as the controls, ± standard error. No statistical significance was found, and (**D**) SEM micrographs of vancomycin-loaded PLA–HA scaffolds by the different methodologies: dip coating (**a**,**b**), drop coating (**c**,**d**) and 3D printing (**e**,**f**), after 6 days of incubation at two magnifications: 200× (**a**,**c**,**e**) and 1600× (**b**,**d**,**f**).

**Figure 3 polymers-15-04250-f003:**
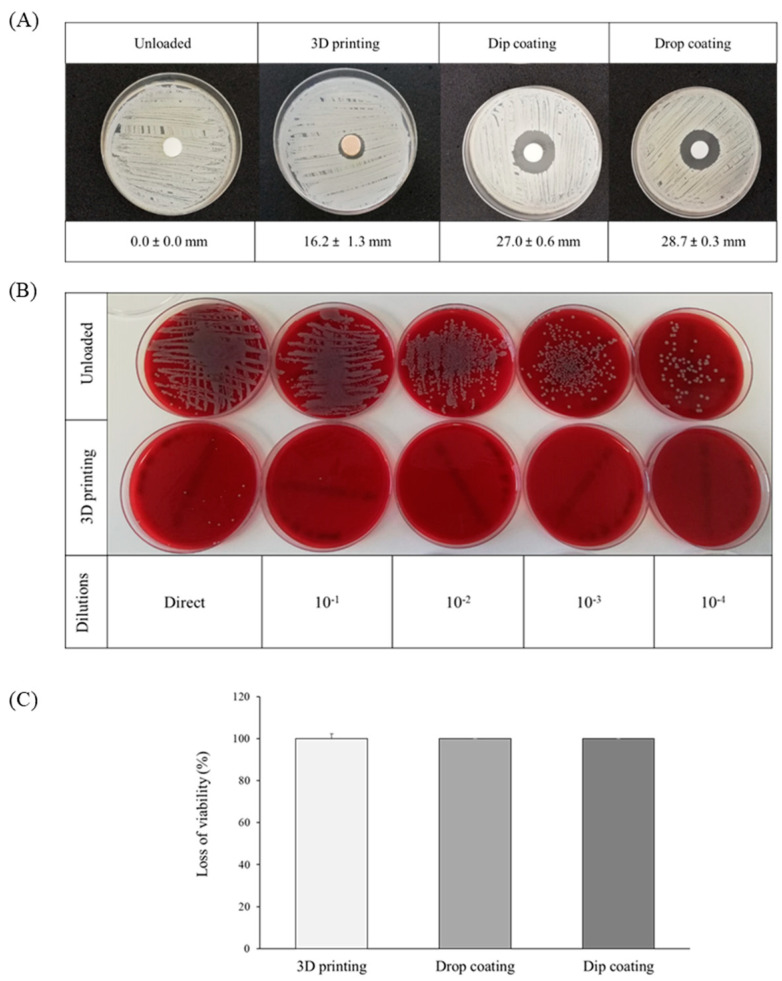
(**A**) Images of the inhibition halos (mm) at 24 h promoted by the antibacterial activity of the scaffolds loaded with vancomycin using the three loading methodologies in Staphylococcus aureus ATCC 2593 cultures on Mueller–Hinton agar plates. The control (unloaded scaffold) is included. (**B**) Images of *S. aureus* growth on agar plates after 24 h using the contact method for unloaded (control) and vancomycin-loaded scaffolds by 3D printing following [33] and (**C**) viability loss (LV%) of *S. aureus* on direct contact with the three types of vancomycin-loaded scaffolds evaluated. No statistical significances were found.

**Figure 4 polymers-15-04250-f004:**
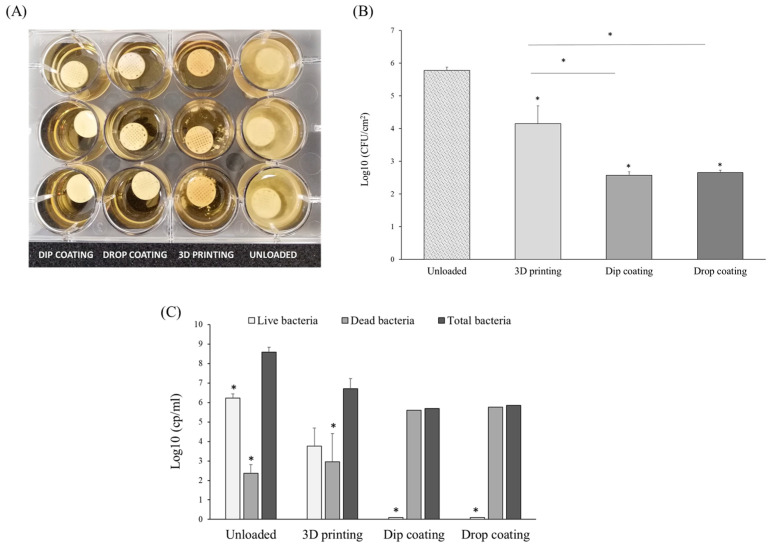
(**A**) Images of *S. aureus* biofilm formation after 48 h of incubation of unloaded (control) and vancomycin-loaded 3D PLA–HA scaffolds by the three methodologies. (**B**) Quantitative evaluation of log reduction CFU/cm^2^ for the unloaded (control) and loaded scaffolds. Statistical significance represented as * for *p* ≤ 0.05. (**C**) Logarithmic reduction in biofilm formation in cp/mL for vancomycin-loaded scaffolds after 48 h of incubation with *S. aureus*. Unloaded scaffolds were included as controls. Statistical significance represented as * for *p* ≤ 0.05 with respect to total bacteria.

## Data Availability

Not applicable.

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
