# Peer review of "Vancomycin-Loaded 3D-Printed Polylactic Acid–Hydroxyapatite Scaffolds for Bone Tissue Engineering"

_polymers, 2023, doi:10.3390/polym15214250_

Round 1

Reviewer 1 Report

Comments and Suggestions for Authors

The authors constructed a composite scaffold by 3D printed technology and vancomycin loaded on polylactic acid/hydroxyapatite scaffold. The experimental results showed that the composite scaffold has good antibacterial properties and osteogenic ability. Overall, the composite scaffold has excellent potential for the treatment of infected bone defects,but will still require major modifications before it can be considered for publication.

1.    It is hoped that the authors will strengthen the manuscript writing and improve the readability of this article.

2.    The explanation about 3D printed technology , bone regeneration and bone repair are not sufficient.

3.  The degradation properties of 3D printed scaffolds are an important parameter for applications. The authors did not state the parameters such as degradation rate and degradation time of this composite scaffold in this manuscript. It is hoped that the authors to add the characterization experiments of this composite scaffold regarding the degradation properties.

4.  It is hoped that the authors to indicate the statistical methods used in the analysis of the data and the plotting of the images(e.g., Figure 2)in the study in an appropriate section of the manuscript.

Comments on the Quality of English Language

Minor editing of English language required

Reviewer 2 Report

Comments and Suggestions for Authors

Pérez-Davila et al. aimed to develop vancomycin-loaded 3D printed PLA/HA scaffolds offering local prevention of infections and personalized biodegradable scaffolds with osteointegrative properties. To this Reviewer, the innovation from this work is minimal. A few changes in data presentation and the addition of functional analysis related to bone regeneration would improve the work, but it will not improve the innovation aspect of it.

1.      Make sure to have Figures composed of multiple results, not just a single graph as in Figure 1. Excellent data presentation is critical. My suggestions are:

a.      New Figure 1: Create a schematic figure with the project’s idea and application. It will lead the reader to understand the work easily. Including model design and real pictures of the 3D scaffolds are also recommended.

b.      New Figure 2: Combine current Figure 1 with Figures 2, 3, and 4. It will allow for creating a figure composition that includes Drug delivery, cell viability studies with fibroblasts and osteoblasts, and morphological assessment of attached cells. Cell viability cannot be over 100%. Please change it to metabolic activity.

c.      New Figure 3: Combine current Figure 5 with Figures 6 and 7.

d.      New Figure 4: Combine current Figure 8 with Figure 9.

2.      Perform functional analysis related to bone regeneration, including alkaline phosphatase activity assay, alizarin red assay, and gene expression analysis and immunostaining for (RUNX2, Col 1A, ALP, BMP2, and OCN).

3.      If not testing bone regeneration in vivo, biocompatibility studies with rodents would significantly improve this work.

Comments on the Quality of English Language

No comment on the English.

Round 2

Reviewer 1 Report

Comments and Suggestions for Authors

The authors have addressed all my points. The manuscript has improved significantly. I believe that it can be accepted in the current form.

Author Response

Thank you very much. 

Reviewer 2 Report

Comments and Suggestions for Authors

Page 3: Move Table 1 to supplementary.

Page 13: All elements in the figure should be original. If not original, permission to reuse should be obtained.

Page 13: Move the figure to the Introduction and use it as Figure 1.

Page 7: Combine Figure 1 with Figure 2 on Page 9.

Author Response

Thank you very much, all the indications have been made in the new version attached. As for some elements or images in figure 1, they are free images that have been properly attributed ("designed by Freepik") as indicated in the source website (https://support.freepik.com/s/article/Press-articles?language=en_US).